# Dissolution Kinetics of International Simple Glass and Formation of Secondary Phases at Very High Surface Area to Solution Ratio in Young Cement Water

**DOI:** 10.3390/ma14051254

**Published:** 2021-03-06

**Authors:** Karine Ferrand, Martina Klinkenberg, Sébastien Caes, Jenna Poonoosamy, Wouter Van Renterghem, Juri Barthel, Karel Lemmens, Dirk Bosbach, Felix Brandt

**Affiliations:** 1Institute for Environment, Health and Safety, SCK CEN, B-2400 Mol, Belgium; sebastien.caes@sckcen.be (S.C.); karel.lemmens@sckcen.be (K.L.); 2Institute of Energy and Climate Research (IEK-6), Nuclear Waste Management and Reactor Safety, Forschungszentrum Jülich GmbH, 52425 Jülich, Germany; m.klinkenberg@fz-juelich.de (M.K.); j.poonoosamy@fz-juelich.de (J.P.); d.bosbach@fz-juelich.de (D.B.); f.brandt@fz-juelich.de (F.B.); 3Institute for Microstructural and Non-Destructive Analysis, SCK CEN, B-2400 Mol, Belgium; wouter.van.renterghem@sckcen.be; 4Ernst Ruska-Centre (ER-C 2), Forschungszentrum Jülich GmbH, 52425 Jülich, Germany; ju.barthel@fz-juelich.de

**Keywords:** ISG glass, high surface area, secondary phases, zeolite, CSH, TEM, thermodynamic predictions, dissolution, high pH, young cement water

## Abstract

Static dissolution experiments were carried out with the reference International Simple Glass under hyperalkaline pH at 70 °C and very high SA/V ratio. Three aspects of glass dissolution behavior were investigated, (1) the rate drop regime and the residual rate (stage II), (2) the formation of secondary phases including thermodynamic aspects, and (3) the microstructure of the interface of altered glass and secondary phases. A very low residual rate of 6 × 10^−6^ g/m^2^d was determined based on boron release, which was several orders of magnitude lower than the initial rate established between the start of the experiments and the first sampling on day 59. The presence of a porous layer with a thickness varying between 80 nm and 250 nm and a pore size between 10 nm and 50 nm was observed. CSH phases with a low Ca/Si ratio of 0.3–0.4 and zeolites were also visible at the surface of the altered glass grains, but no glass alteration resumption occurred, probably due to an important pH decrease already at day 59. Thermodynamic calculations assuming congruent glass dissolution and precipitation of the dissolved aqueous species confirmed the precipitation of CSH phases and zeolites.

## 1. Introduction

Immobilization of high- and intermediate-level nuclear wastes by vitrification is a well-established process that has been extensively studied in many countries [1]. Borosilicate glasses are commonly used as a matrix for nuclear wastes due to their flexibility with regard to waste loading and their capacity to incorporate many waste elements, combined with a good glass-forming ability, chemical durability, mechanical integrity, and a good thermal and radiation stability [2,3]. For the storage of the vitrified nuclear wastes, there is a consensus between the waste agencies of many countries that vitrified nuclear wastes should be disposed of in a deep geological waste repository. Different international concepts have been proposed on which cement will be present, e.g., as lining, for plugs, or as part of the general construction of the repository [4,5,6]. In the case of the Belgian Supercontainer and the U.K. disposal concepts for the storage of high- and intermediate-level vitrified wastes, the presence of cement is expected to play a significant role. Specific studies on the effect of Ordinary Portland Cement or portlandite on the dissolution behavior of high- or intermediate-level vitrified wastes have been conducted in Belgium (Supercontainer design), the U.K. (colocation of different waste types), and France (intermediate-level vitrified wastes) [7,8,9].

In contact with water, borosilicate glasses are metastable and dissolve. In static dissolution experiments, often a surface alteration layer (SAL) forms on the dissolving glass. Some tracer elements, e.g., boron or lithium, are assumed to be retained neither in the SAL nor in secondary phases. Based on their release rates, three stages of glass dissolution are distinguished as a function of the reaction progress: (I) initial dissolution, described by a congruent glass dissolution at the highest rate, (II) residual dissolution, characterized by a glass dissolution rate several orders of magnitude lower than the initial one, and (III) resumption of glass alteration, which is typically observed under specific conditions, i.e., high temperature and/or high pH [10,11]. Microscopically, the formation of a complex SAL has been identified as a prerequisite for the slower dissolution kinetics of stage II. In many cases, electron microscopy revealed that the SAL consists of a silica-based layer made of aggregated colloids and sometimes exhibits a layered structure [9,12,13]. Different glass dissolution models explaining the mechanism of the SAL formation and the rate-limiting steps have been proposed and are still under debate. Most of the glass dissolution experiments in the literature were performed under near-neutral pH conditions and at 90 °C, following standard procedures (e.g., MCC-1 static leach tests or Product Consistency Tests), and did not show any glass alteration resumption (stage III). However, in many studies under alkaline conditions, such an alteration resumption was observed due to the formation of secondary phases [10,14,15,16,17,18]. To determine the long-term stability of the waste glasses in a cementitious environment, it is important to know in which conditions (pH, temperature, solution composition) a residual rate is reached and maintained without passing to stage III.

The classical glass dissolution model includes the formation of the SAL via solid-state diffusion of protons into the glass combined with the diffusion of cations out of the glass [19,20]. More recently, high-resolution techniques such as Transmission Electron Microscopy (TEM), nanoscale Secondary Ion Mass Spectrometry (nanoSIMS), or Atom-Probe Tomography (APT), as well as in situ Raman measurements during glass alteration, have provided new insights [21,22]. Some of the resulting observations, e.g., a sharp boundary in the element concentrations between the SAL and the pristine glass, challenged the original idea of the SAL as a residual and restructured glass and introduced the process of a continuous complete dissolution followed by precipitation to describe glass alteration [12,23,24]. New models were thus developed based on the microscopic and spectroscopic observations, including congruent initial glass dissolution, formation of a Si-rich interfacial solution layer, amorphous silica precipitation, formation of the SAL, and diffusive transport through the SAL [22,25]. These models consider an interdiffusion process between protons and glass network species inside the glass as well as a SAL composed of amorphous silica. Regardless of the mechanism proposed for the SAL formation, a passivating effect of this layer was considered [19,22,25,26,27]. Under hyperalkaline conditions, a dissolution-reprecipitation process was suggested as the mechanism to form the SAL [11,18] which was seen as a diffusion barrier with a protective effect in some studies, e.g., [18]. The formation of secondary phases can then lead to a destabilization of this protective layer by consuming its main constitutive elements.

Apart from the glass composition, the following parameters were identified to have a significant impact on the formation of secondary phases and the extent of glass alteration resumption: temperature, glass surface area to solution volume ratio (SA/V), pH, and cations in solution [9,28]. Experiments at alkaline pH with a complex solution composition, e.g., with the addition of Ca in cementitious water, indicate a strong dependence of glass alteration kinetics on the formation of secondary phases such as clay minerals, calcium silicate hydrates (CSH), or zeolites [18,29]. Especially the formation of zeolites can significantly enhance the dissolution kinetics compared to stage II [11,28,30]. In some cases, the formation of CSH phases in addition to zeolites was linked to an increase in the glass dissolution rate [31]. In a recent study in which International Simple Glass (ISG) was altered in young cement water (YCWCa) at 70 °C and a SA/V of 8280 m^−1^, the formation of a very thick SAL (>15 µm) was observed on which secondary phases identified as zeolite and phyllosilicate were formed [9]. However, neither the formation of a CSH phase nor a glass alteration resumption occurred.

The aim of the investigations presented in this paper was to study the combined effect of hyperalkaline conditions and very high SA/V ratio on the dissolution of ISG and the formation of secondary phases. Experiments were carried out at 70 °C in a synthetic young cement water containing Ca (YCWCa), with experimental conditions similar to those selected in [9], but with a much higher SA/V ratio, i.e., 264,000 m^−1^ instead of 8280 m^−1^. This enables a direct comparison and allows assessing the effect of the SA/V ratio on the formation of secondary phases, the structure and composition of the SAL, and the glass dissolution kinetics. Starting from a series of independent glass dissolution experiments, we applied a combined macroscopic and microscopic approach to determine the glass dissolution kinetics and to characterize the glass surface alteration and the formation of secondary phases. The identification of secondary phases was achieved by applying the combination of a grain size fractionation and the enrichment of secondary phases in particular fractions.

## 2. Materials and Methods

### 2.1. Experimental and Analytical Techniques

Static dissolution experiments were performed under Ar atmosphere at 70 °C in young cement water containing Ca (YCWCa), with the composition given in Table 1. The composition of YCWCa was calculated considering exchange with the Boom Clay water [32].

YCWCa was prepared in a glove box under Ar atmosphere according to the procedure described in [33]. The pH was measured at room temperature using a WTW 340 pH meter (Xylen Analytics, Weilheim in Oberbayern, Germany) combined with a Metrohm micro-electrode (InLab Micro, Mettler Toledo, Greifensee, Switzerland) calibrated with independent buffers. A pH of 13.7 ± 0.2 was measured at room temperature, corresponding to pH 12.5 at 70 °C by considering the water dissociation constant (K_w_) increase with temperature increase, i.e., K_w_ = 1.58 × 10^−13^ at 70 °C.

For the dissolution experiments, a very fine powder was prepared from the International Simple Glass (ISG) produced by MoSci Corporation (Rolla, MO, USA, Lot number: L12012601-M12042403) with the following nominal composition in wt.%: 56.2% SiO_2_, 17.3% B_2_O_3_, 12.2% Na_2_O, 6.1% Al_2_O_3_, 5.0% CaO, and 3.3% ZrO_2_ [11]. The glass block was cut into smaller pieces, milled, and sieved to a size fraction between 20 and 25 µm. The powder was then cleaned ultrasonically in water for a few minutes to remove the ultrafine fraction. This treatment does not induce glass dissolution features like etch pits or porosity. The specific surface area of this powder, measured by multipoint BET (ASAP 2020, Micromeritics, Norcross, GA, USA) using Kr adsorbent gas, was equal to 0.440 ± 0.002 m^2^/g. The pristine glass powder observed by Scanning Electron Microscopy (SEM, Phenom PRO X, Thermo Fisher Scientific, Breda, The Netherlands) is shown in Figure 1.

For each dissolution experiment, 3 g of ISG powder was placed in contact with 5 g of YCWCa, resulting in a very high SA/V ratio of 264,000 m^−1^. The weight of the Teflon^®^ containers was checked at the end of the respective experiment to correct the data for possible evaporation. A stir bar coated with Teflon^®^ was introduced in the container for manual stirring and solution homogenization once a week. Each container corresponds to a different sampling time; i.e., one cell was dismantled after day 59, 288, 385, 632, and 952, respectively. The experimental conditions are summarized in Table 2.

At the end of each experiment, the solution was collected using a needle attached to a syringe, pH was measured at room temperature, and the solution was filtered through a 0.45 µm membrane filter. Then, 1 mL of solution was diluted with 2 mL Milli-Q water (18.2 MΩ cm at 25 °C) and ultrafiltered (10 kD) at 5000 rotations per minutes (RPM) for 20 min. To determine the elemental concentrations by ICP-OES (IRIS Intrepid II dualview, Thermo Scientific, Waltham, MA, USA) or ICP-MS (XSERIES2, Thermo Scientific, Waltham, MA, USA), the sample was then diluted and acidified with either 5% HCl (Merck nv, Overijse, Belgium) and 1% HNO_3_ (Thermo Fisher GmbH, Kandel, Germany) or 1% HNO_3_, respectively.

A powder sample was taken after 59, 288, and 385 days for microscopic analysis of the complete powder, i.e., for the observation of the SAL and the secondary phases formed on the glass. In order to collect the altered glass powder, the solution was first removed using a syringe with a needle and treated as previously described. Then, to rinse the powder, milli-Q water was added and immediately removed also using a syringe with a needle. The powder was dried under inert atmosphere in an oven at 30 °C until a stable weight was reached. Samples of this complete altered glass powder were directly analyzed with X-ray Diffraction (XRD), Scanning Electron Microscopy (SEM), and Energy Dispersive X-ray Spectroscopy (EDS). XRD analysis was performed with a D4 diffractometer (tD4, Bruker, Karlsruhe, Germany) with copper Kα radiation at 40 mA and 40 kV. SEM-EDS characterization was carried out with a Phenom apparatus scanning electron microscope (SEM, Phenom PRO X, Thermo Fisher Scientific, Breda, The Netherlands) using an accelerating voltage of either 5 kV or 15 kV, and with an environmental SEM Quanta 200 FEG (FEI, Eindhoven, The Netherlands) combined with Energy Dispersive X-ray Spectrometry (Apollo X SSD, EDAX, Weiterstadt, Germany)) in a low vacuum mode.

Subsamples with a size fraction lower than 5 µm were prepared from the complete glass powder altered for 385 days by sedimentation based on Atterberg’s principle. For this size fraction separation, 0.5 g of the altered glass powder was introduced in a 50 mL test tube and filled with Milli-Q water up to a height of 10 cm. The suspension was dispersed by shaking and after a settling time of 1 h and 15 min at 23 °C, 4/5 of the suspension was removed and centrifuged at 4000 RPM for 5 min. The supernatant solution was withdrawn to keep only the altered glass powder with a grain size < 5 µm. This procedure was repeated until the supernatant fluid was clear. To perform XRD measurements, the suspension was deposited on a Si-holder and dried at room temperature. Complementary to XRD, the fraction < 5 µm was also inspected with SEM and secondary phases were analyzed for their chemical composition with EDS. In addition, the grain size fraction > 5 µm and the complete glass powder after 385 days were analyzed via XRD, SEM, and EDS.

Samples for Transmission Electron Microscopy (TEM) were prepared by focused ion beam milling (FIB). In preparation of the measurements, some of the complete dried altered glass powder was embedded in an epoxy resin (EpoFix). This embedded sample was then polished with diamond suspensions (9, 3, and 1 µm, respectively) and examined by SEM to select a representative particle for a detailed TEM investigation. A thin lamella was prepared using an NVision 40 cross beam station (NVision 40, Carl Zeiss AG, Oberkochen, Germany) (Appendix A), as described in [22]. The cross-section lamella was observed with a JEOL 3010 transmission electron microscope (JEOL, Japan) operating at 300 kV. Conventional bright-field images were recorded showing mainly absorption contrast. EDS spot measurements (Link-ISIS, Oxford Instruments, Abingdon, UK) were carried out to determine the local composition.

Additional investigations with TEM and Scanning Transmission Electron Microscopy (STEM) were carried out with an FEI Tecnai G 2 F20 (Thermo Fischer Scientific, Breda, the Netherlands) operated at 200 kV accelerating voltage [34]. High-angle annular dark-field (HAADF) images were recorded in STEM mode realizing Z-contrast imaging of the thin cross-section [35]. For the STEM observations, the electron probe was formed with 12 mrad convergence semi-angle, and the intensities of diffracted beams were recorded with an annular detector covering the angular range of 45 mrad up to about 200 mrad in a magnified image of the back-focal plane. In this HAADF setup, the recorded intensity increases with increasing accumulated atomic core charge at the position of the small scanning probe. Higher intensity indicates a composition with a higher atomic number, larger sample thickness, or higher material density. The TEM bright field image contrast is inverse to the STEM HAADF contrast. Energy-dispersive X-ray Spectroscopy (EDS, INCA-system Oxford instruments, Oxford, UK) was applied for the qualitative determination of the SAL composition.

### 2.2. Calculations of Normalized Mass Loss and Dissolution Rate

The normalized mass loss NL_i_ (g/m^2^) was calculated according to Equation (1):(1)$$\;NL_{i}=\frac{\left(C_{i}\;\cdot \;V\cdot \;F_{i}\;\right)}{f_{i}\;\cdot \;SA}=\frac{\left(m_{i}\;\cdot \;F_{i}\right)}{\left(f_{i}\;.\;\frac{\;SA}{V}\right)}$$
where *C_i_* is the concentration of element *i* in the aliquot of solution (mg/L), *V* is the total volume of solution (m^3^), *F_i_* is the factor to convert the atomic weight of element *i* to the molecular weight of the oxide-containing element *i, f_i_* is the weight % of the oxide-containing element *i* in the pristine glass, *SA* is the total surface area of the exposed glass (m^2^), and *m_i_* is the mass of element *i* (g). Note that for the calculation of the NL_Na_ and NL_Si_, the respective Na and Si concentrations initially present in YCWCa were subtracted from the measured concentrations. The glass dissolution rate (r) was derived from the normalized mass loss of boron, NL_B_, according to Equation (2):r = d(NL_B_)/dt(2)

### 2.3. Thermodynamic Modeling of Secondary Phases Formation

The reaction pathway of ISG dissolution in YCWCa and formation of secondary phases towards equilibrium was modeled with the GEM-Selektor code package [36] using the thermodynamic database Thermodem [37], which contains thermodynamic data on cementitious phases and zeolites from [38,39,40]. In GEM-Selektor, the equilibrium state is determined via direct minimization of the total Gibbs energy of the system, defined by its bulk elemental composition, temperature, pressure, and standard Gibbs energy per mole of all chemical species. The activity model used in the calculation follows the extended Debye-Hückel model for aqueous electrolytes. The typical input recipe for the modeled system contained 1 kg of YCWCa, according to the chemical composition specified in Section 2.1, and 600 g of ISG. Calculations were performed for T = 70 °C and P = 1 bar.

The thermodynamic calculations presented here were based on the assumption of congruent dissolution and formation of secondary phases according to [11] and based on the dissolution-reprecipitation model developed in [12,23]. The reaction pathway towards equilibrium was obtained by calculating the partial equilibrium assuming a congruent dissolution of 80, 106, 113, 150, 175, 200, and 600 g of glass in 1 kg of YCWCa. This method allows the determination of the secondary phases that are thermodynamically likely to form and thus provides a qualitative understanding of the evolution of the system, based on the amount of glass that has dissolved. The dissolution of 106 and 113 g of glass corresponds to the total amount of glass dissolved at days 288 and 385 of the experiments, based on the calculated NL_B_. The formation of high-temperature phases (i.e., microcline and grossular), unlikely under the current experimental conditions, was excluded in the calculations.

## 3. Results

### 3.1. Elemental Concentrations and pH Evolution, Initial and Residual Dissolution Rates

The elemental concentrations of B, Na, Si, K, Ca, and Al and the measured pH with time are summarized in Appendix A. The concentrations of B and Na increased steeply already at the beginning of the experiments and remained stable after day 59. B concentrations of about 8000 mg/kg were measured, which were significantly higher than in the previous experiments at SA/V of 8280 m^−1^, where the maximum B concentrations were about 5000 mg/kg. Hence, a higher reaction progress was reached. The decrease in the B and Na concentration between 288 and 632 days is due to the evaporation of the part of the solution in the first period (up to 385 days), leading to a higher concentration in the leachate. It is also important to remember that each data point is the result of another experiment, which leads to a supplementary variation of the resulting leachate concentrations. In contrast, after a sharp increase until day 59, the concentration of Si continuously decreased until reaching a stable concentration of about 580 mg/kg after day 632. The concentration of Ca, which is present in YCWCa, decreased from 17.8 mg/kg at the start of the experiments to a minimum of 2 mg/kg, and then increased again up to a steady-state concentration of about 7 mg/kg after day 385. The concentrations of K (only present in YCWCa) and Si (coming from the glass) behaved similarly and decreased to low, constant concentrations after day 288. For Al, very low concentrations were measured, i.e., <0.12 mg/kg after 59 days and little further increase. The main change in the pH value was already observed at the first sampling point; on day 59, the pH_(70 °C)_ already dropped from 12.5 to 9.8. Then, pH remained constant for the duration of the experiments, with a final pH_(70 °C)_ of 9.4 at day 952 (Figure 2). A resumption of glass dissolution was not observed during the time scale of the experimental series.

The normalized mass losses NL_i_ calculated from the elemental concentrations are shown in Figure 2. The NL_i_ evolutions of B and Na do not show the decrease that was seen for the evolution of the concentrations as a result of evaporation, suggesting that the evaporation did not have much effect on the dissolution kinetics (Appendix A).

NL_i_ indicates the formation of a SAL and/or secondary phases at the very early stage of the experiments. Already at day 59, all NL_i_ were lower than NL_B_, with very low NL_i_ for Ca, Zr, and Al throughout the complete experimental series. Most similar to boron was the evolution of Na, but even NL_Na_ remained well below NL_B_ at all observed experimental times, suggesting that Na was also involved in secondary phase formation, and is thus not a good glass dissolution tracer in these experiments.

Based on NL_B_, an initial dissolution rate of 6 × 10^−3^ g/m^2^d was calculated between 0 and 59 days and a final rate of 6 × 10^−6^ g/m^2^d was calculated between 288 and 952 days. Because the final rate is very low, it is considered a residual rate in stage II of the ISG dissolution. The experiment after 385 days was chosen for further investigations of the solid as it already represents stage II of ISG dissolution.

### 3.2. Thermodynamic Predictions of Stable Secondary Phases

Figure 3 shows the secondary phases, assuming congruent glass dissolution and precipitation of new phases until the experiment reaches equilibrium. The calculations are based on the total boron release for a given reaction progress. The calculations indicate the stable phases quartz (SiO_2_), okenite (Ca[Si_2_O_5_]∙2H_2_O, a crystalline CSH phase), and zeolites (phillipsite-Na and phillipsite-K, i.e., NaAlSi_3_O_8_∙2H_2_O and KAlSi_3_O_8_∙2H_2_O, respectively)_._ At thermodynamic equilibrium, the leached Zr from the glass is expected to form baddeleyite (ZrO_2_) as the most stable phase.

Table 3 summarizes the calculated amounts of secondary phases formed and the calculated total aqueous ion concentrations in solution corresponding to an experimental time of 385 days; for comparison, the measured solution composition is also given. The calculated aqueous concentrations of Na, B, and SO_4_ are comparable to the measured solution composition. However, there are significant differences between the measured and calculated compositions of the aqueous phase with respect to K, Ca, and Si.

### 3.3. Evolution of Secondary Phases with Time: SEM Observations of Altered Glass from Day 59 and 288

Samples of the early stages of dissolution were taken on days 59 and 288 for SEM inspection. Already at the early stages of the dissolution experiments, secondary phases were observed. Figure 4 shows representative particles of the altered glass, including newly formed crystals (arrows). These had the classical morphology of zeolites and increased in their typical size from day 59 to day 288 and became more idiomorphic by developing characteristic faces and twinning. Furthermore, the surfaces of the altered glass particles appeared rough because of a surface covering, which is characterized in detail later on.

### 3.4. Secondary Phases after 385 Days: Detailed Characterization by XRD and SEM-EDS

The complete glass powder and the grain size fractions >5 µm and <5 µm altered for 385 days were analyzed in detail by XRD and SEM-EDS as these samples were considered representative for stage II of dissolution. The XRD pattern of the complete altered glass as well as of the grain size fraction >5 µm exhibited a very broad peak characteristic for amorphous silica or poorly crystallized silicates (Figure 5a). Some diffraction peaks were, however, detected. Additional peaks of crystalline phases appeared in the grain size fraction <5 µm. The identification of the crystalline phases on this fine grained powder fraction showed calcite (PDF 01-083-0578), a Na-phillipsite (PDF 01-074-1787), and a K-zeolite (PDF 01-085-0976; Figure 5b).

On the complete altered powder, SEM-EDS analyses showed three different secondary phases (Figure 6a, Appendix A). This sample contained (1) very fine, fibrous particles covering the glass grain surfaces; (2) larger, elongated particles of a sodium-aluminum silicate originating from the glass surface; (3) large, idiomorphic zeolites that are often twinned. Detailed EDS characterization and images of each secondary phase are shown in Appendix A. Based on EDS analyses, the presence of zeolites was confirmed (Appendix A). The zeolites appeared to have grown directly on the glass surfaces as observed in the complete glass sample (Figure 6a).

The sample treatment and grain size separation efficiently removed the fine secondary phases that covered the glass surface with a carpet-like layer in the complete altered glass powder. After the removal of the secondary phases, the altered glass particles of the size fraction >5 µm (Figure 6b) exhibited smooth glass grain surfaces. Furthermore, round pores of different sizes became more clearly noticeable on the glass grain surfaces, some of them exhibiting halos, i.e., smaller pores within larger pores. Not all glass particles are altered to the same extent. Some particles are completely covered with pores, while other ones look unaltered. The grain size fraction > 5 µm also contained some large idiomorphic crystals of zeolite, which appear damaged by the sample treatment (Figure 6b). The grain size fraction < 5 µm consisted of very fine-grained particles, small glass fragments, and fibrous crystals. The presence of calcite was confirmed in this fraction based on the chemical composition of particles with a typical rhomboedric morphology (Figure 6c, Appendix A).

An EDS elemental mapping was performed on a sample of the complete altered glass powder, which was embedded and polished as described in Section 2.1 (Figure 7). The BSE image (Figure 7a) shows pores, mainly on the outer rim and also on the surfaces of the glass particles, as previously observed in Figure 4 and Figure 6. Secondary phases attached to the glass surface were observed in this sample as well. On the scale of SEM-EDS, the distribution of Al in the glass particles was homogeneous, whereas the presence of a zeolite caused a local Al enrichment (arrow in Figure 7). Around the glass particles, a thin zone of significantly different chemical composition was detected, corresponding to the carpet-like layer of secondary phases observed on individual glass grains in Figure 4. In this zone, an enrichment of K was observed as well as in some of the round pores (large glass particle in the lower left corner in Figure 7). Na was homogeneously distributed in the glass, was detected not enriched in the layer of secondary phases, and was not present at all in the zeolite.

### 3.5. Glass and Secondary Phases Interface Characterization by TEM-EDS

Complementary to SEM-EDS observations, a detailed investigation was made at the interface of glass and secondary phases on a FIB lamella, which was prepared from a representative, embedded altered glass particle taken from the experiment after 385 days (Figure 8, Appendix A). The FIB section covered an area starting from the glass, including a very thin zone of different contrast (marked with arrows in Figure 8), to the secondary phases (the fibrous layer previously mentioned) which were loosely attached to the particle surface (Figure 8). In contrast to experiments with lower SA/V ratios (e.g., [9,22]), no typical SAL with a colloidal structure was detected in this sample. An additional feature within the FIB lamella was a large pore (1 µm in diameter), which was filled with a fine-grained phase (Figure 8, Figure 9b).

The pristine glass appeared homogeneous (left part of the lamella, Figure 9a). Within the error of the method, TEM-EDS spot analyses of the pristine glass were in good agreement with the theoretical glass composition. Appendix A of the supplementary information summarizes all measurements carried out in the different areas of this lamella. Close to the pristine glass surface, a thin porous altered layer was observed. The thickness of the porous altered layer varied between 80 nm and 250 nm and the pore size between 10 nm and 50 nm (Figure 9c,d). The pores were empty and appeared not to be connected. Compared to the pristine glass composition, the porous altered layer showed a lower content of Na and Ca, whereas K and Si were enriched (Appendix A).

The secondary phases attached to the porous altered layer exhibited a fibrous morphology typical for CSH phases. These fibers formed a loose layer with a thickness between 0.5 µm and 1.5 µm. In comparison to the pristine glass, their chemical composition showed an increase in Ca and K, a decrease in Si, but no Al and Zr (Appendix A). The Ca/Si ratio in these fibrous phases was 0.3–0.4, suggesting the formation of CSH with a low Ca/Si ratio. Due to their low crystallinity and instability towards the electron beam, the detection of an electron diffraction pattern was not possible. In addition, the fine-grained phase in the large pore (Figure 9b) was analyzed by TEM-EDS. The Al, Si, and K contents of this phase were similar to those of the porous altered layer, but it was enriched in Na and depleted in Ca and Zr. The Al/Si ratio of the pristine glass, the porous altered layer, and the fine-grained phase in the large pore was constant with a value of 0.12.

STEM-HAADF investigations allowed for a more detailed structural and chemical view at the very delicate zone of porous altered layer and secondary phases (Figure 10). In the detailed observations depicted in Figure 10, no local chemical gradient was observed between the adjacent glass and the region with pores. This might indicate that the porous altered layer did not represent a newly formed phase but rather a region attacked by dissolution. In the overview image of Figure 10, the CSH phases appeared to have grown out of the pores and might therefore be responsible for the pore formation.

STEM EDS mapping shown in Figure 11 highlights the presence of some areas in which the chemical composition was different, i.e., mainly enriched in Si and Ca. This might suggest that the CSH phases were restructured to more stable forms as predicted by thermodynamic calculations.

## 4. Discussion

The dissolution of ISG glass in the complex YCWCa solution at a very high SA/V ratio and 70 °C was investigated in several independent static dissolution experiments. Three aspects of glass dissolution were addressed in this study, (1) the rate drop regime and the residual rate of stage II, (2) the formation of secondary phases including thermodynamic aspects, and (3) the microstructure of the interfaces of pristine glass–altered glass and altered glass–secondary phases.

The initial dissolution rate of ISG established between 0 and 59 days equals 6 × 10^−3^ g/m^2^d, which is six times lower than the forward dissolution rate, i.e., the highest rate of stage I, determined at pH = 13.5 and 30 °C in YCWCa by performing dynamic experiments [41]. Due to a faster network hydrolysis with increasing temperature, the forward dissolution rate at 70 °C is expected to be higher than the one measured by Elia et al. (2017) [41]. From the equation established by [19] and taking the fitting parameters from [42] resulting from the modeling of ISG dissolution experiments at alkaline pH, a forward dissolution rate of 11 g/m^2^d was calculated at 70 °C. Our initial dissolution rate is thus several orders of magnitude lower than this calculated forward dissolution rate, due to a significant drop in pH between day 59 and the beginning of the experiment.

In the final stage (stage II) of ISG dissolution, i.e., between 59 and 952 days, the residual dissolution rate of 6 × 10^−6^ g/m^2^d is several orders of magnitude lower than the initial rate. Such a dissolution behavior is expected based on the general dissolution model for alumino-borosilicate glasses [19,43,44,45]. The residual dissolution rate in our study is 150 times lower than the one reported in [9] for ISG alteration at 70 °C in YCWCa, but at a much lower SA/V ratio. The difference observed in the residual rates can be explained by the different pH evolution in both dissolution experiments. Indeed, in the study of Mann et al. (2019) [9], the pH, measured at room temperature, slowly dropped from 13.5 at the beginning of the test to pH_(RT)_ around 11.7 after 200 days. Later pH values oscillated around 12, corresponding to variations in the porosity/density of the SAL observed by TEM. In the experimental series presented here, already after 59 days, the pH_(70 °C)_ dropped to 9.8, i.e., to 11.2 at RT. Then, it steadily decreased by small increments to a final value of 9.4 at 70 °C, i.e., to 10.8 at RT. No pH oscillations occurred, indicating a continuous, steady dissolution regime during stage II of glass dissolution. Dissolution proceeds at a very low rate, especially considering the relatively high temperature. In comparison to available literature values for glass dissolution at high pH, the residual ISG dissolution rate in our study is very similar to the residual dissolution rate determined in [46] for a synthetic basaltic glass altered at 90 °C and high SA/V ratio of 10,000 m^−1^ if our dissolution rate is converted to a geometric surface area normalized rate as described in [46]. Although the temperature in this latter study was higher, the final pH was similar to the one measured in our experiments, as a pH value of 9.4 at 70 °C corresponds to around 9 at 90 °C.

Thermodynamic modeling predicts the formation of CSH phases (okenite, Table 3). CSH phases were also predicted in the dissolution experiment series with ISG and YCWCa at a lower SA/V ratio of [9], but not experimentally observed. One reason for this could be that at a low SA/V ratio, the silicon required to form CSH phases precipitated as silica spheres in the multiple bands of a thick SAL formed on ISG, whereas the faster release of glass constituents into the solution at a high SA/V ratio fostered the direct formation of the more stable CSH phases. Especially in the first 59 days of the experiment, the use of a very high SA/V ratio allowed higher concentrations of all elements involved in the formation of CSH and zeolites to be reached, so CSH phases quickly covered the surfaces of the glass particles.

The final pH, the occurrence of zeolites, and CSH followed the thermodynamic prediction. No quartz or other crystalline or amorphous SiO_2_ was observed; i.e., no enrichment of Si was visible in the EDX-mapping (Appendix A). An additional phase, which was not predicted by thermodynamic calculations for the early stages of glass dissolution, was Na-phillipsite, which acts as a sink for Na and Si. The evolution of the Na-normalized mass loss (NL_Na_) was different from NL_B_ due to this secondary phase formation, and therefore Na could not be used as a good glass dissolution tracer. The precipitated calcium carbonates are likely to be a drying artifact, as previously observed on samples from glass dissolution experiments at high pH [9,47].

The main differences between calculated thermodynamic equilibrium and the measured aqueous composition concern K, Ca, and Si. This might partly be due to the formation of amorphous to nano-crystalline CSH phases or calcium aluminum silicate hydrate (CASH) phases instead of the predicted precipitation of the crystalline Ca-silicate okenite. These CSH and CASH phases can also take up alkalis like K into their interlayer [48], decreasing the aqueous concentration of K. The formation of K-rich phases would also lead to a further pH decrease, as observed experimentally. If the formation of the thermodynamically most stable phase, e.g., okenite, is suppressed, a mineral of higher solubility, e.g., tobermorite, will precipitate, increasing the aqueous Si and Ca concentrations. This suggests that the formation of the crystalline Ca-silicates (okenite or tobermorite) is in fact kinetically hindered. Instead, CSH phases of higher solubility with a lower Ca/Si ratio are kinetically favored. Therefore, these precipitated as observed experimentally. In contrast to Ca, the predicted excess Si is not observed. This can be explained by the presence of an additional Na-phillipsite. Based on our calculations, the leached Zr in aqueous solution is likely to form baddeleyite. However, this phase was not observed experimentally. Kinetic difficulties of attaining the equilibrium have been previously reported for zirconia when the equilibrium was approached from oversaturation [49,50].

Zeolite and CSH formation have been reported in many studies on glass dissolution at high pH and were connected to glass alteration resumption [10,11,28,47]. Further sensitive analysis of our thermodynamic calculations showed that if the formation of quartz is inhibited, other less crystalline SiO_2_ phases (chalcedony, or christobalite) are likely to occur. However, this would have no impact on the amount of zeolite and CSH. On the other hand, suppressing the formation of baddeleyite would trigger the formation of amorphous Zr(OH)_4_. This would have an impact on the distribution of Si, Al, Ca, and Na in the minerals and enable the additional formation of Na-Phillipsite.

In our experiments, the presence of the secondary phases formed at a very high SA/V ratio did not lead to a dissolution resumption. The reason for this could be the decreasing pH. In some previous studies, it was demonstrated that, in free-pH experiments, a threshold pH around 10.5 at 90 °C exists, at which the precipitation of zeolites becomes very slow or even no longer occurs [15,28]. According to the thermodynamic calculations, zeolites and CSH are all stable throughout the duration of the experiments presented here. Judging from the SEM images taken on days 59 and 288, zeolites grow in number and size with time. This may be due to Ostwald ripening or to kinetic effects favoring zeolite growth versus CSH growth at lower pH. The CSH phases seem to be present at day 59 and not to grow significantly after day 288. Since the CSH phases are covering the glass surface directly, the glass dissolution may be linked to their growth kinetics; i.e., without a significant growth of the CSH phases, glass dissolution slows down.

After removal of the secondary phases from the altered glass grains, some glass particle surfaces appear to be inhomogeneously attacked by dissolution with areas of preferential dissolution, causing the formation of pores in the surfaces. Earlier studies at alkaline pH also showed the formation of pitted features on ISG altered at alkaline pH as well [51]. They attributed these to preferential localized dissolution at sites with a higher alkali concentration or to a separate, less durable, vitreous phase.

Very few studies so far have provided a detailed microscopic view of the interface between glass and secondary phases after dissolution at high pH and high SA/V ratio. In contrast to many experiments presented in the literature carried out at low SA/V ratio [12,13,22], no colloidal SAL was observed. TEM observations only showed a very fine porous altered layer of 80–250 nm at the interface between the pristine glass and the secondary phases. Its microstructure is similar to the one formed during the dissolution of synthetic basaltic glasses at a high SA/V ratio, which consists of a foam-like structure instead of colloids [46]. However, this latter study also described secondary phases inside the pores, whereas these contained no solid phase in the altered ISG observed in our study [46]. An average amount of dissolved glass can be derived from the boron concentration data by dividing NL_B_ after 385 days with the theoretical ISG density of 2.51 × 10^6^ g/m^3^. The calculated alteration layer thickness of 168 nm is in good agreement with the TEM observation of the porous altered layer. This also confirms that born is a good glass dissolution tracer for ISG at the chosen experimental conditions. In glass dissolution experiments at very alkaline conditions, the precipitation of calcium borates was only observed by [52], who used Ca(OH)_2_ solution as alteration solution. Such phases were not identified in the studies [9] and [18], in which ISG was altered in YCWCa, i.e., in the same medium as in our study.

Based on the combination of macroscopic and microscopic observations, ISG dissolution at high pH and very high SA/V ratio can be interpreted as a dissolution-reprecipitation process, as proposed by many authors regarding glass alteration under extreme pH conditions [12,13,18]. In [11], due to a much lower SA/V ratio, the glass dissolution rate, as well as the pH, stayed high. In our study, the drop in pH slowed down the growth of CSH phases and zeolites as well as the glass dissolution itself. The presence of the thin porous altered layer at the interface pristine glass and altered glass hinders the direct contact between pristine glass and aqueous solution. However, due to the unknown properties of the pore filling, a protective effect of this layer can only be postulated.

## 5. Conclusions

The combined effect of hyperalkaline conditions and very high SA/V ratio on the dissolution rate of ISG, the formation of an altered glass surface and secondary phases was studied at 70 °C in a synthetic YCWCa. The present results show that the SA/V ratio is a key parameter for the dissolution rate and for the formation of the altered glass surface and secondary phases. In good agreement with the general trends of alumino-silicate glass dissolution, the initial dissolution rate established between the start of the experiments and 59 days was several orders of magnitude higher than the residual dissolution rate of 6 × 10^−6^ g/m^2^d.

The combination of a relatively high temperature (70 °C), a synthetic cementitious water containing Na, K, and Ca at pH 13.5, and a very high SA/V ratio led to the formation of secondary phases identified as CSH phases and zeolites by SEM, XRD or TEM, XRD or TEM. Thermodynamic blind predictions assuming congruent glass dissolution and precipitation of the dissolved aqueous species indicated the formation of quartz, okenite (CSH), phillipsite-Na and phillipsite-K (zeolites), and baddeleyite as the most stable phases. However, in our dissolution experiments, the formation of the crystalline Ca-silicates was kinetically hindered, leading to the formation of metastable CSH phases with a lower Ca/Si ratio of 0.3–0.4. The precipitation of baddeleyite and quartz was also not observed experimentally, probably due to kinetic difficulties to reaching equilibrium. In summary, stable phases such as K-zeolite were present as well as metastable Na-zeolite and CSH of kinetically controlled composition. Contrary to dissolution experiments under very alkaline conditions and at low SA/V ratio, the precipitation of secondary phases at very high SA/V ratio did not lead to a glass alteration resumption. This is ascribed to a drop in pH before the onset of the residual dissolution regime.

The microstructure of the altered glass surface consisted of a very thin, porous layer from which CSH phases appeared to have grown. Instead of a typical gel-like colloidal morphology, the porous layer was made of a foam-like structure with voids that contained no solid phases. Such a microstructure was already described earlier for the dissolution of other glass types at a very high SA/V ratio as well. In combination with the pH drop, the newly formed altered glass surface might also contribute to low initial and residual dissolution rates.

In the context of the Belgian deep geological disposal for the vitrified high-level waste, high pH near the field due to the use of concrete potentially has a large effect on the glass dissolution rate. The expected pH evolution (decrease) in the near field will be favorable and may lead to low long-term glass dissolution rates, but a better understanding of the dissolution and transport mechanisms is necessary to quantify the impact and the time scale. Our dissolution tests in the absence of cement at very high reaction progress showed that an important pH drop can occur, leading to residual glass dissolution rates as in neutral pH conditions. This pH decrease depends much on the type of phases that are formed. A kinetic dissolution model coupled with precipitation of secondary phases will be applied to relate the amount of dissolved glass to the evolution of the solution’s pH.

## Figures and Tables

**Figure 1 materials-14-01254-f001:**
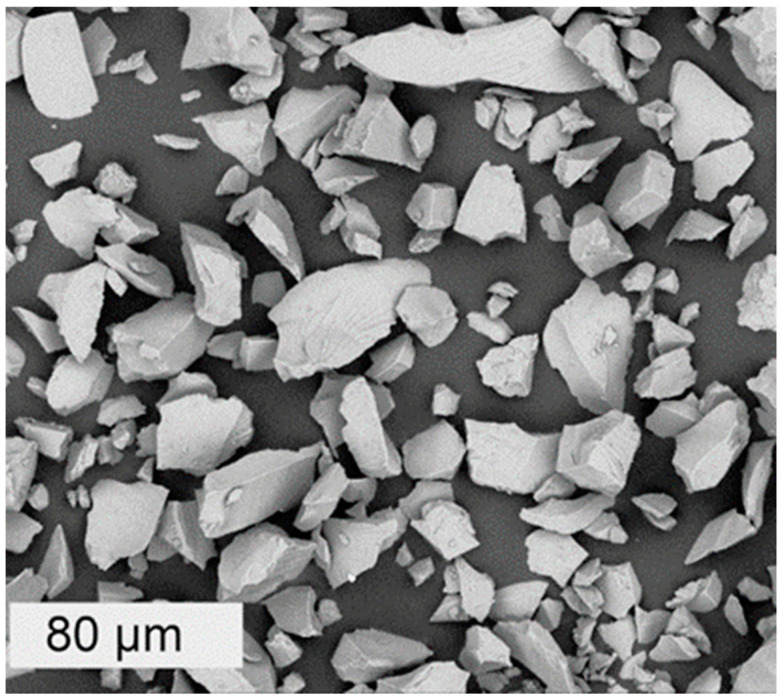
SEM micrograph of the pristine International Simple Glass (ISG) powder used in the dissolution experiments.

**Figure 2 materials-14-01254-f002:**
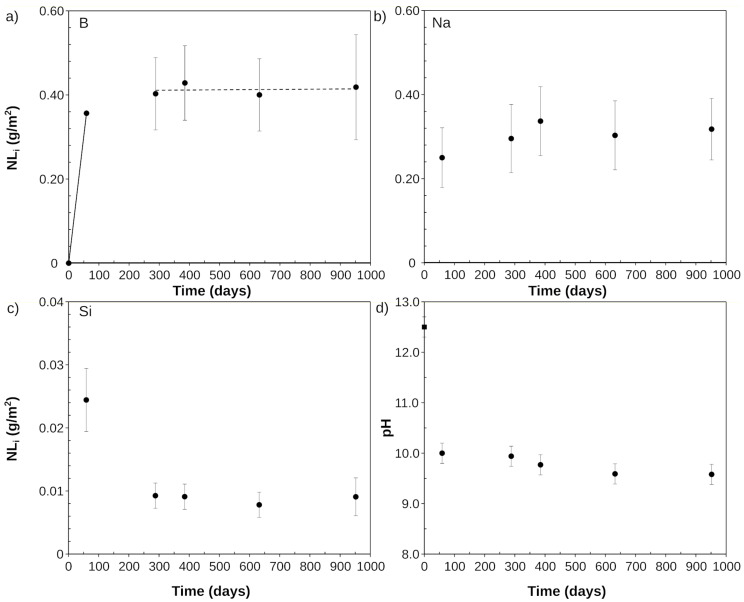
Normalized mass loss NL_i_ of ISG components up to 952 days in YCWCa at 70 °C: (**a**) NL_B_ (showing the linear regressions from which the dissolution rates were derived), (**b**) NL_Na_ and (**c**) NL_Si_; (**d**) evolution of the pH_(70 °C)_ with time.

**Figure 3 materials-14-01254-f003:**
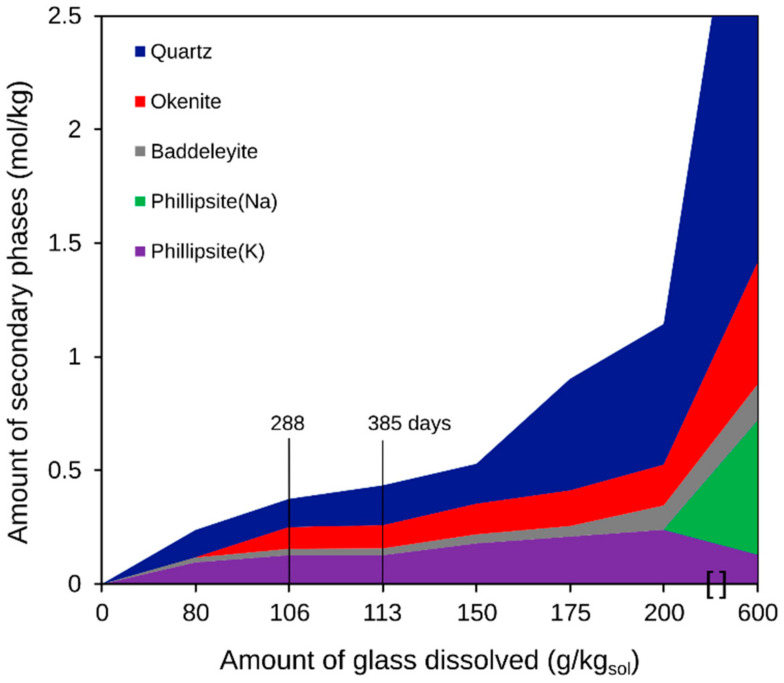
Predicted secondary phases formed following a sequential congruent glass dissolution up to 600 g of ISG in 1 kg of YCWCa. The amounts of 106 g and 113 g of dissolved glass correspond to the solution composition after 288 and 385 days in the dissolution experiments, respectively.

**Figure 4 materials-14-01254-f004:**
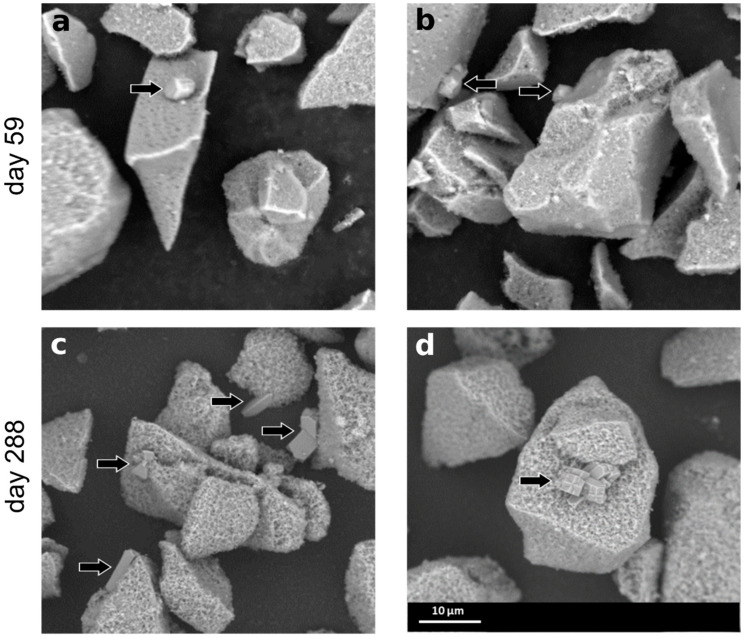
SEM-SE images of altered glass samples after 59 (**a**,**b**) and 288 days (**c**,**d**).

**Figure 5 materials-14-01254-f005:**
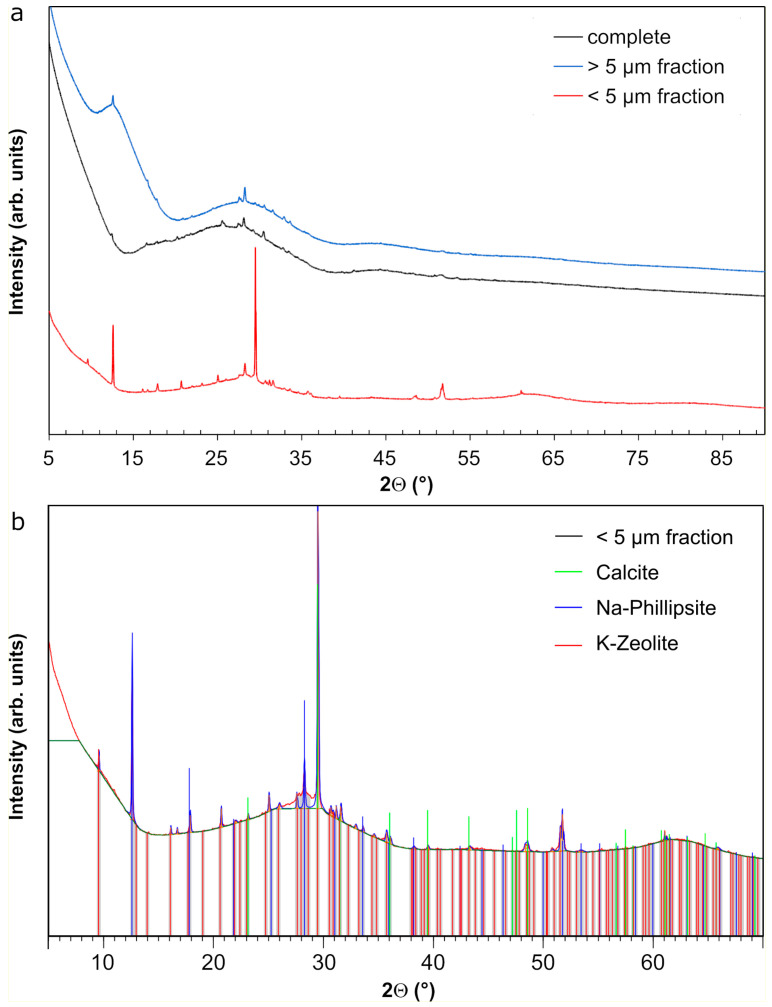
(**a**) XRD patterns of the complete glass powder altered for 385 days in YCWCa at 70 °C, and of the grain size fractions <5 µm and >5 µm. (**b**) Zoomed in XRD pattern of the grain size fraction <5 µm with the identified crystalline phases and their respective peak positions.

**Figure 6 materials-14-01254-f006:**
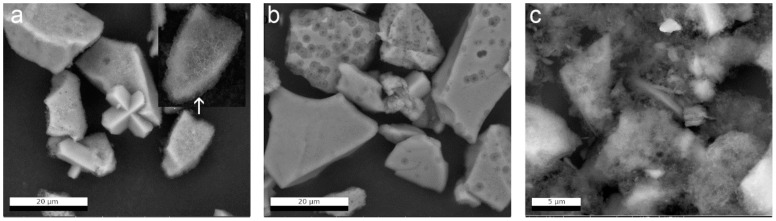
SEM-BSE image of (**a**) the complete altered glass powder after 385 days in YCWCa at 70 °C, i.e., before grain size fractionation, (**b**) the grain size fraction >5 µm, (**c**) the grain size fraction <5 µm.

**Figure 7 materials-14-01254-f007:**
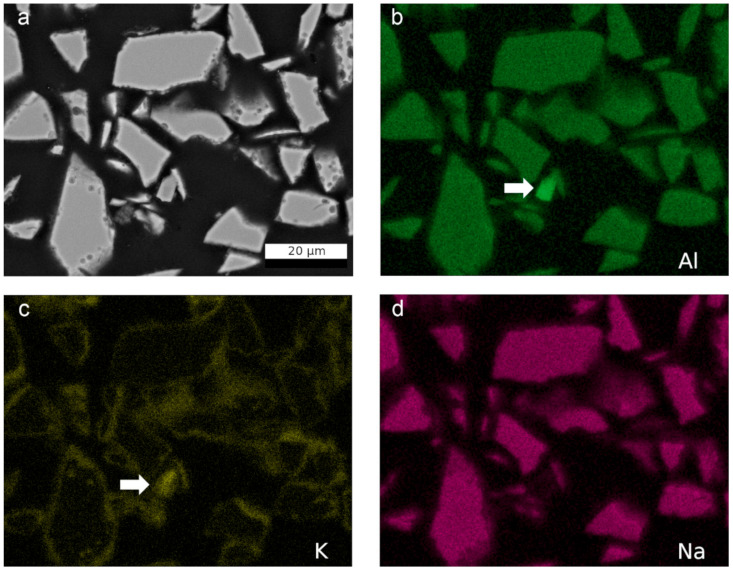
SEM-BSE image (**a**) and corresponding elemental mappings of Al-K lines (**b**), K-K lines (**c**), and Na-K lines (**d**) of embedded altered glass taken after 385 days in YCWCa at 70 °C.

**Figure 8 materials-14-01254-f008:**
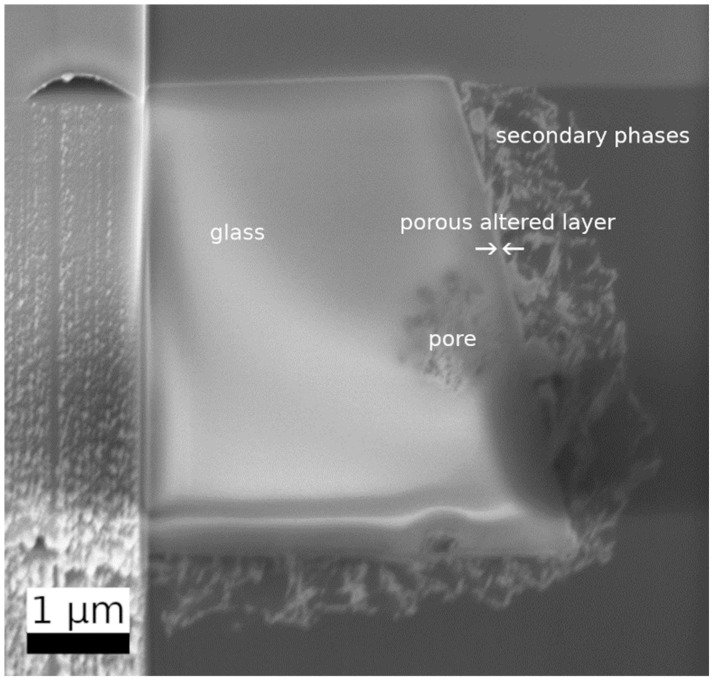
SEM in-lens secondary electron image of the thinned glass grain with the dimensions 2.9 µm (top), 4.2 µm (bottom), height 4.6 µm. The pristine glass was surrounded by a very thin altered layer (between the arrows) on which an about 1 µm thick layer of fibrous secondary phases was formed.

**Figure 9 materials-14-01254-f009:**
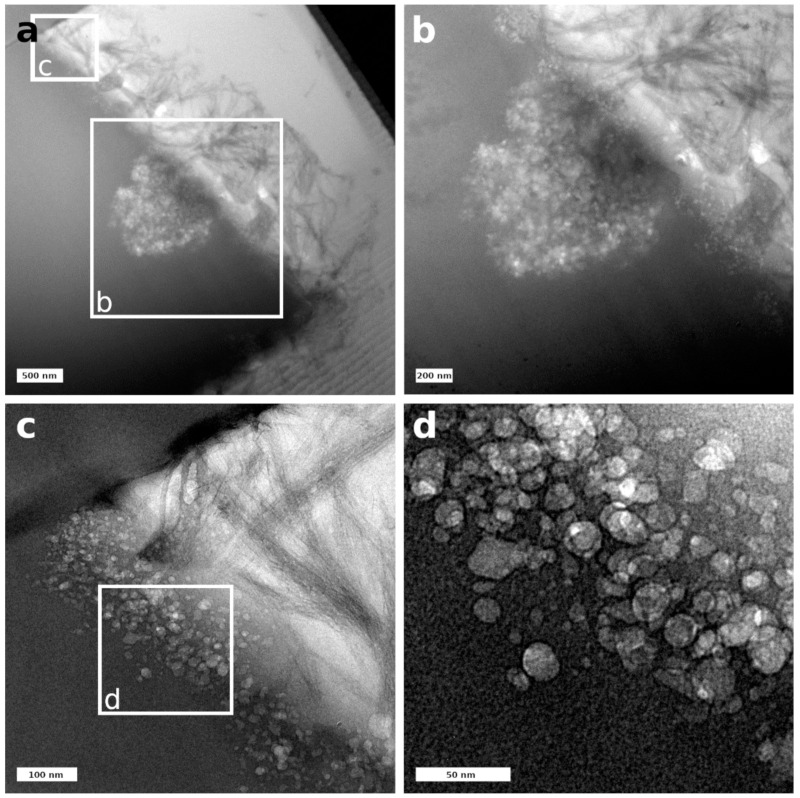
(**a**) Overview of the pristine glass–porous altered layer interface, (**b**) close-up of the large pore (image a), (**c**) close-up of the pristine glass–porous altered layer interface (image a), (**d**) close-up of pores in the altered layer (image c).

**Figure 10 materials-14-01254-f010:**
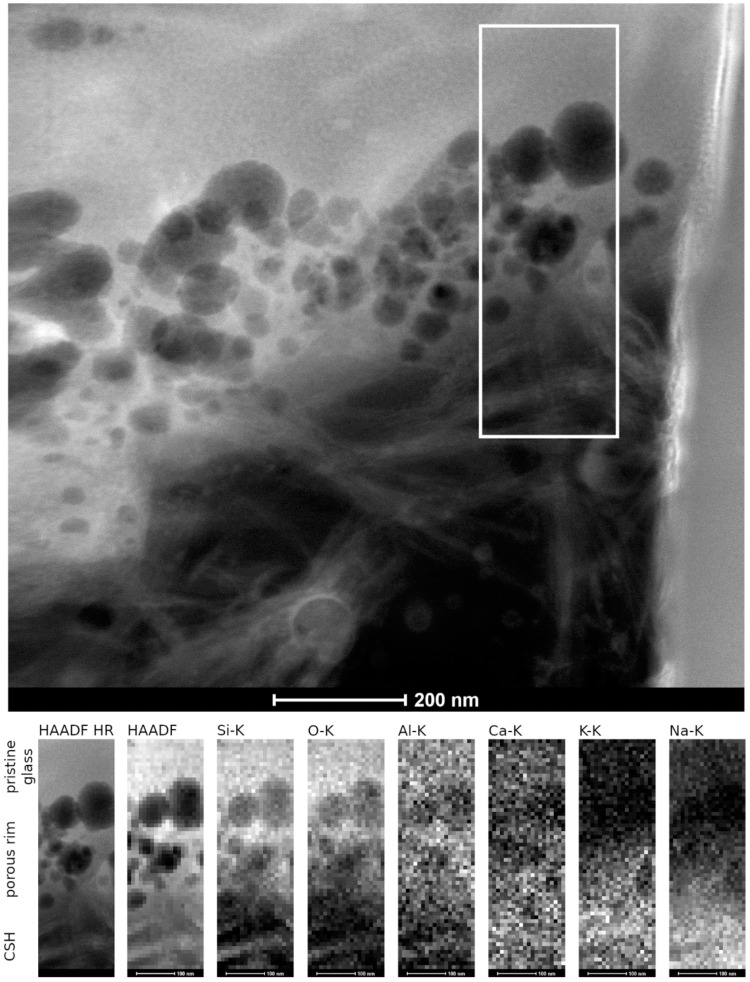
STEM EDS mapping of the porous altered layer–secondary phases interface as observed in the HAADF STEM image. Elemental mappings of Si-, O-, Al-, Ca-, K-, and N- K lines.

**Figure 11 materials-14-01254-f011:**
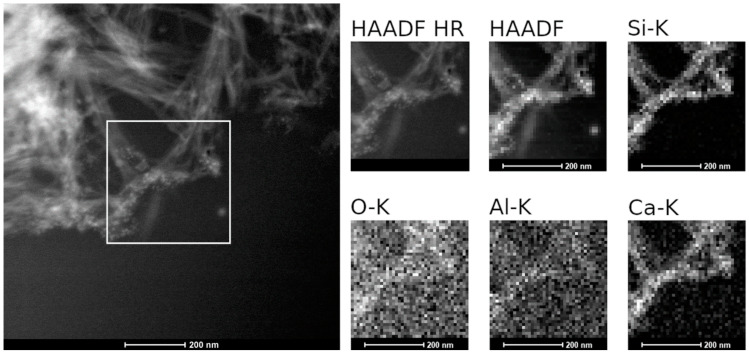
STEM EDS mapping: overview of CSH phases and selected area for EDS mapping. Elemental mappings of Si-, O-, Al-, and Ca-K lines.

**Table 1 materials-14-01254-t001:** Content of elements in the young cement water containing Ca measured by ICP-OES.

Element	Al	B	Ca	Na	K	Si
Concentration (mg/kg)	0.06 ± 0.04	<1	17.8 ± 1.8	3120 ± 310	12,400 ± 1200	0.48 ± 0.21

**Table 2 materials-14-01254-t002:** Experimental conditions of the static glass dissolution experiments at very high SA/V ratio.

Parameter	Setting
Temperature (°C)	70
Particle fraction (µm)	20–25
Mass of glass powder (g)	3 ± 0.005
Specific surface area of glass powder by BET (m^2^/g)	0.440 ± 0.002
Solution composition	YCWCa, pH (70 °C) = 12.5 ± 0.2
Weight of solution (g)	5 ± 0.005
SA/V (m^−1^)	264,000
Duration (days)	59, 288, 385, 632, 952

**Table 3 materials-14-01254-t003:** Calculated amounts of secondary phases and calculated and measured solution composition after 385 days in YCWCa at 70 °C.

Secondary Phase/Solution Composition	Calculated (113 g Glass Dissolved)	Experiment (385 Days)
Phillipsite (K) (mol/kg_sol_)	0.18	-
Okenite (mol/kg_sol_)	0.10	-
Quartz (mol/kg_sol_)	0.18	-
Na (mg/L)	13,856	12,807
Al (mg/L)	<0.01	0.12
Ca (mg/L)	0.34	7.83
K (mg/L)	8911	4549
Si (mg/L)	1164	723.75
Zr (mg/L)	0.00	0.0006
B (mg/L)	6356	6969
SO_4_ (mg/L)	196	188
pH_(70 °C)_	10.1	9.6

## Data Availability

Data is contained within the article or Appendix A.

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
