# Peer review of "Dissolution Kinetics of International Simple Glass and Formation of Secondary Phases at Very High Surface Area to Solution Ratio in Young Cement Water"

_materials, 2021, doi:10.3390/ma14051254_

Round 1

Reviewer 1 Report

Comments to the authors:

This is a very interesting work and a great number of tests and discussion have been done for this research.

- Everywhere through the text, “oC” should be attached to numbers. For example, “70oC” not “70 oC”.

- What is “stage II” in line 3 of the abstract. This needs to be defined before use it in abstract.

- Why the static dissolution experiments were carried out at 70oC?

- SEM analysis of Figure 1 should be presented in the text.

- The effective parameters to control the secondary phase and should be discussed in conclusion section.

Best Regards,

Reviewer

Reviewer 2 Report

The paper reports the results of combined effect of hyperalkaline conditions and very high SA/V ratio on the dissolution of International Simple Glass and the formation of secondary phases. Authors concluded that the SA/V ratio is a key parameter for the dissolution rate and for the formation of the altered glass surface and secondary phases. They found a very low residual dissolution rate which was several orders of magnitude lower than the initial rate. Authors have also found that the microstructure of the altered glass surface consisted of a very thin, porous layer and that instead of a typical gel-like colloidal morphology, the porous layer was made of a foam-like structure with voids which contained no solid phases. The paper provides important data to be shared with the scientific community. It is well-written and is of commendable quality being recommended to publication with following comments to be accounted for:

First, the Table 1 is not “Composition of the YCWCa synthesized…” as claimed, it is the content of elements in the young cement water containing Ca.

Second, Line 135 informs readers that “The powder was then cleaned ultrasonically in water to remove the ultrafine fraction.”. Could authors comment on changes that could occur in the glass during its cleaning in water? Can they be fully neglected? Are there any estimations of near-surface changes due to the contact of glass with the water? It is known that real nuclear waste glass samples on contact with water show in the very initial period of contact with water a short-term fast leaching of species from glasses termed instantaneous surface dissolution – see the equation (24) of reference "Corrosion of alkali-borosilicate waste glass K-26 in non-saturated conditions", J. Nucl. Mat. 340, 12-24 (2005). This term is particularly important for the opposite case when SA/V ratio is very low, and this is most probably the case of glass samples washing. How this short-term fast leaching of species from glasses is accounted if glass washing is done using water? 

Author Response

"Please see the attachement"

Reviewer 3 Report

This paper is devoted to a rather lengthy and useful study of the interaction of glass with water, which simultaneously interacts with cement, and therefore properties of the water change significantly. This is a useful and great study. However, I have several comments and recommendations.

1. It is not clear why the authors chose such time intervals for their research (59, 288, 385, 632, and 952 days)? How can this be explained? Why is there such a long interval between 1 and 2 points? Judging by Figure 2, the most interesting changes occur between points 1 and 2, but unfortunately, we do not see their details.

2. It would be beneficial to the paper if the authors supplemented the conclusion with a couple of sentences about how useful the results they obtained can bring to the process that stimulated their research (immobilization of nuclear waste). They received new data. What follows from this data for this process? Maybe change it to make it better? How?
